energy/synthetic chemistry

amphoteric copolymer, dispersant, coal water slurry, free radical polymerization, adsorption

**Author for correspondence:**
Guanghua Zhang
e-mail: zhanggh@sust.edu.cn

# Synthesis of a novel amphoteric copolymer and its application as a dispersant for coal water slurry preparation

Lun Du[1], Guanghua Zhang[1], Dongdong Yang[1], Jie Luo[1], Yewei Liu[2], Wanbin Zhang[3], Ce Zhang[3], Junguo Li[1] and Junfeng Zhu[1]

[1]Shaanxi Key Laboratory of Chemical Additives for Industry, Shaanxi University of Science and Technology, Xi'an 710021, People's Republic of China
[2]China Coal Technology and Engineering Group Clean Energy Co Ltd, Beijing 100013, People's Republic of China
[3]Shaanxi Collaborative Innovation Center of Industrial Auxiliary Chemistry and Technology, Shaanxi University of Science and Technology, Xi'an 710021, People's Republic of China

LD, 0000-0002-8890-3428; GZ, 0000-0003-1260-5245

In this work, a novel amphoteric copolymer named Poly(sodium p-styrenesulfonate–*co*-acrylic acid-*co*-diallyldimethylammonium chloride) (P(SS-*co*-AA-*co*-DMDAAC)) was synthesized via free radical polymerization. Afterwards, P(SS-*co*-AA-*co*-DMDAAC) was explored for use as a dispersant in coal water slurry (CWS) preparation. The structure of P(SS-*co*-AA-*co*-DMDAAC) was verified by Fourier transform infrared spectroscopy and nuclear magnetic resonance. The synthetic conditions were optimized as the feed ratio of AA to SS was 1 : 1 (for Yulin coal) or 1.5 : 1 (for Yili coal), and DMDAAC dosage was 4.0 wt% (for Yulin coal) and 6.0 wt% (for Yili coal) toward total monomers. The performances of P(SS-*co*-AA-*co*-DMDAAC) as a dispersant for CWS were evaluated by various technologies, such as apparent viscosity, zeta potential, static stability and contact angle measurements. The results revealed that the optimized dosage of P(SS-*co*-AA-*co*-DMDAAC) in CWS preparation was 0.3 and 0.4 wt% for Yulin coal and Yili coal respectively. In this optimum condition, CWS prepared using P(SS-*co*-AA-*co*-DMDAAC) as dispersant showed a typical shear thinning behaviour and excellent stability, which are desired in industries. The rheological models also confirmed the pseudo-plastic characteristics of CWS. Finally, compared with the widely used anionic dispersant naphthalene sulphonate formaldehyde condensate (NSF) and poly(sodium

p-styrenesulfonate) (PSS), P(SS-*co*-AA-*co*-DMDAAC) developed in this work exhibited better slurry making performance. The introduction of cationic functional groups promoted the adsorption of the dispersant, which further enhanced the electrostatic repulsion and steric hindrance among coal particles. Accordingly, the viscosity of CWS decreased and static stability enhanced.

# 1. Introduction

Over the past decades, amphoteric copolymers have attracted great attention in science and industry areas due to their special physical and chemical properties. A number of amphoteric copolymers with different chemical composition have been designed and synthesized for used as dispersant, stabilizer, flocculant and superplasticizer [1–5]. Coal water slurry (CWS), which is a clean coal-based fuel and raw material for the coal-to-chemicals industry, is usually composed of pulverized coal, water and a small amount of dispersant. In practical industry, CWS with a high coal load, low viscosity and high stability is in demand. Therefore, a small amount of dispersant is critically important for the dispersion and stabilization of coal particles in CWS system.

The most commonly used dispersants can be categorized into natural compounds and synthetic polymers according to their origins. Natural compound dispersants such as lignin [6–8] and humic acid derivatives [9,10] possess advantages such as low cost, abundant reserve and environment friendliness. However, they usually exhibit poor dispersion performance, and large-scale applications have been limited. Naphthalene sulfonic formaldehyde condensate (NSF), as a synthetic dispersant, exhibits improved dispersion performance [11–13]. However, their synthetic route usually demands harsh conditions and causes environmental pollution. In recent years, copolymers that simultaneously bear various functional groups (such as carboxyl and sulfonic groups) have been developed for use as a novel dispersant for CWS. For instance, various polycarboxylate dispersants with different structures and chemical composition have been reported [14–16].

According to established theory, the dispersant works by adsorbing on the coal particles surface and further improve the surface properties. Therefore, the adsorption process has significant influences on the dispersion performance of dispersant. In general, the adsorption of dispersant on coal particles is mainly driven by hydrophobic interaction between the dispersant and coal surface, such as the adsorption of NSF [12,13]. However, the composition of coal surface is complex: except for the hydrophobic area, a partial positively charged area is also present on the coal surface because of cations in ash [17], and a partial negatively charged area exists due to the ionization of oxygen functional groups [18–20]. In most cases, the coal surface is negatively charged as a whole because oxygen group content is much higher than ash content. In accordance with this fact, the introduction of cationic functional groups into the structure of dispersant is beneficial to the adsorption of dispersant. However, the relevant reports are still limited. Recently, our group [21] synthesized a dispersant bearing cationic groups by using methacryloyloxy ethyl trimethylammonium chloride (DMC) as monomer. As expected, the performances of this dispersant were encouraging. However, the cationic groups in DMC units were relatively far away from the main chain of the copolymer, which may not be conducive to the formation of closer adsorption between the dispersant and the coal surface.

In this work, a novel copolymer named poly(sodium p-styrenesulfonate–*co*-acrylic acid-*co*-diallyldimethylammonium chloride) (abbreviated as P(SS-*co*-AA-*co*-DMDAAC)), which contains both cationic and anionic functional groups was designed and synthesized. To enhance the adsorption of dispersant, the anionic groups (–COOH) of the copolymer were used to combine with the cations in ash by complexation, and the cationic groups (N$^+$ groups) could combine with the negative oxygen group by electrostatic attraction. Thus, not only the hydrophobic area, but also the positively charged and negatively charged area could act as adsorption site for the dispersant. Meanwhile, the $SO_3^-$ groups provided the electrostatic repulsion force among coal particles. The structure of the copolymer was characterized by Fourier transform infrared spectroscopy (FTIR) and nuclear magnetic resonance (NMR). Afterwards, the synthetic conditions of P(SS-*co*-AA-*co*-DMDAAC) were optimized according to the viscosity of CWS prepared by using Yili coal and Yulin coal respectively. Finally, the performances of P(SS-*co*-AA-*co*-DMDAAC) as a dispersant in CWS were studied systematically. The slurry making performance of P(SS-*co*-AA-*co*-DMDAAC) was compared with that of the widely used anionic dispersant NSF and PSS. And the adsorption and dispersion mechanism of P(SS-*co*-AA-*co*-DMDAAC) was proposed.

**Table 1.** Proximate and ultimate analysis.

| sample | proximate analysis (wt%) | | | ultimate analysis (wt%) | | | | |
|---|---|---|---|---|---|---|---|---|
| | $M_{ad}$ | $A_{ad}$ | $V_{daf}$ | $C_{ad}$ | $H_{ad}$ | $N_{ad}$ | $O_{ad}$ | $S_{ad}$ |
| Yili | 11.50 | 15.20 | 40.53 | 68.80 | 4.27 | 0.81 | 15.80 | 0.32 |
| Yulin | 5.71 | 9.89 | 34.18 | 70.96 | 4.34 | 0.99 | 10.79 | 0.71 |

# 2. Material and methods

## 2.1. Materials

Acrylic acid (AA, 99%) was obtained from Fuchen Chemical. Sodium p-styrenesulfonate (SS, 90%) and diallyldimethylammonium chloride (DMDAAC, 60% water solution) were purchased from Aladdin Reagent Co., Ltd. Sodium bisulphite ($NaHSO_3$) and ammonium persulfate (($NH_4$)$_2S_2O_8$) were purchased from Tianjin Tianli Chemical Reagent Co., Ltd. Sodium hydroxide (NaOH) was provided by Tianjin Damao Chemical Reagent Co., Ltd.

## 2.2. Preparation

### 2.2.1. Synthesis of amphoteric copolymer

P(SS-*co*-AA-*co*-DMDAAC) was synthesized by convenient free radical polymerization technology, which is beneficial for large-scale production. The typical synthesis process is described as follows. SS, $NaHSO_3$ and deionized water were added to a flask under stirring. After SS was dissolved thoroughly, the mixture was warmed to 80°C. Then, AA, DMDAAC and ($NH_4$)$_2S_2O_8$ aqueous solutions were simultaneously dripped into the flask within 1.5 h and the mixture was allowed to react for another 2 h. After polymerization, the product was cooled to room temperature. The pH value of the solution was adjusted to 7–8 by using 1 M NaOH solution. The crude product solution was concentrated, and further purified through precipitation in an excess of ethyl alcohol. Finally, P(SS-*co*-AA-*co*-DMDAAC) was obtained as a faint yellow powder.

### 2.2.2. Preparation of coal water slurry

To reliably evaluate the performances of P(SS-*co*-AA-*co*-DMDAAC) for used as a dispersant in CWS, two types of coal from Yili (Xinjiang Province, China) and Yulin (Shaanxi Province, China) were adopted for the preparation of CWS, respectively. The proximate and ultimate analyses of the coal are listed in table 1. The coal was pulverized by a FM-3 pulverizer (Shanghai Ke Heng Industrial Co., Ltd., China). The coal particles with different particle size distribution were mixed based on the following ratio: 25 wt% of 124–420 µm, and 75 wt% of less than 74 µm coal particles [22,23]. The coal was mixed with a calculated amount of dispersant and water. After stirring for 10 min at 600 r.p.m., CWS was obtained and subjected to further measurements.

## 2.3. Measurements

### 2.3.1. FTIR spectrometry

The FTIR spectrum was measured by using a VECTOR-22 FT-IR instrument (Bruker, Germany). Approximately 5 mg copolymer sample was mixed with 500 mg KBr in a mortar, and then compressed into a disc. The scanning was performed from 4000 to 500 cm$^{-1}$.

### 2.3.2. $^1$H NMR spectrometry

The $^1$H NMR spectrum was obtained on an AVANCE NMR spectrometer (400 MHz, Bruker, USA). D$_2$O was used as the solvent and the measurement was conducted at 295.5 K.

### 2.3.3. Measurements of adsorption amount of dispersants on coal surface

The adsorption capacity of dispersant was analysed by using a Cary-60 ultraviolet spectrophotometer (Agilent, USA). A series of dispersant solutions were prepared and measured to build the calibration

curve. Then, 20 ml dispersant solutions of different concentration were mixed with 0.5 g blend coal sample and vibrated in a water bath at 25°C for 5 h. The suspension was centrifuged at 8000 r.p.m. for 10 min, and the supernatant was filtered to remove coal particles. The concentration of residue dispersant in the supernatant was measured by ultraviolet spectrophotometer. Coal-water suspensions were used as the blank sample. The adsorption amount of dispersant was calculated by following equation (2.1).

$$\Gamma = \frac{(C_o - C_r + C_{blank}) \times V}{m},\tag{2.1}$$

where $\Gamma$ (mg g$^{-1}$) is adsorbed dispersant on unit mass coal sample, $C_o$ (mg ml$^{-1}$) is original mass concentration of dispersant solution, $C_r$ (mg ml$^{-1}$) is residue mass concentration of dispersant solution after adsorption, $C_{blank}$ (mg ml$^{-1}$) is the mass concentration of blank sample, $V$ (ml) is the volume of dispersant solution, $m$ (g) is the mass weight of coal sample.

### 2.3.4. Measurements of the adsorption layer thickness formed by dispersants on coal surface

The X-ray photoelectron spectrum (XPS) of the coal samples were obtained by using an AXIS SUPRA spectrometer (Kratos, UK) with a monochromatic Al Kα source of 120 W. The binding energy was corrected based on the C1s peak at 284.6 eV.

Before the XPS measurements, blended coal sample was mixed with dispersant solution and vibrated at 25°C for 5 h. Then the suspension was filtered to remove water and dispersant that had not been adsorbed. After that, the coal sample was dried at 105°C to constant weight and then attached to the sample plate for XPS analysis. Coal is known to contain Si element, while the dispersant has no Si; thus the Si 2p can be used as the character element to estimate the thickness of the adsorption layer formed by the dispersant. The photoelectron intensity will decay after travelling through the adsorption layer. On the other words, the thickness of adsorption layer is negatively related to the photoelectron intensity. On the basis of the method from previous paper [24], the adsorption layer thickness of dispersant that formed on the coal surface was calculated by using the equations (2.2)–(2.4) as follows:

$$d = -\ln\left(\frac{I_d}{I_0}\right) * \lambda(E_k),\tag{2.2}$$

$$\lambda(E_k) = 49E_k^{-2} + 0.11(E_k)^{1/2}\tag{2.3}$$

and

$$E_k = h v - E_b,\tag{2.4}$$

where $I_d$ is the intensity of the photoelectron transmitted through the adsorption layer, $I_0$ is the incident photoelectron intensity, $d$ is the thickness of adsorption layer (nm), $\lambda(E_k)$ is the average depth (nm) at which the light electron escaped, $E_k$ is the light electron kinetic energy and $E_b$ is the atomic binding energy of Si. The value of $I_0$ and $I_d$ can be obtained according to the area of the Si 2p photoelectron in the XPS spectrum.

### 2.3.5. Zeta potential measurements

The zeta potentials of the coal particles were measured by using a NANO-ZS-90 dynamic light scattering instrument (Malvern Instruments Corp., UK). The coal suspension consists of 0.2 g blended coal and 50 ml dispersant solution (0–0.6 mg ml$^{-1}$), which was vibrated in water bath at 25°C for 5 h and then centrifuged at 8000 r.p.m. for 10 min. The supernatant was isolated for zeta potential analysis. Each sample was measured three times and the mean value was used.

### 2.3.6. Contact angle measurements

Before the contact angle measurements, 2.0 g blended coal was compressed to a disc with a diameter of 15 mm and a thickness of 2 mm at 60 MPa for 10 min. Then, distilled water and dispersant solution were dropped onto the surface of the discs respectively. The contact angles were measured by a DSA100 dynamic contact angle measuring instrument (Kruss, Germany).

### 2.3.7. Apparent viscosity and rheology of CWS

The apparent viscosity of CWS samples was tested on a R/S-SST Plus rheometer (Brookfield Company, US). The measurements were carried out at 25°C. The average viscosity at a shear rate of 100 s$^{-1}$ was counted as the apparent viscosity of CWS.

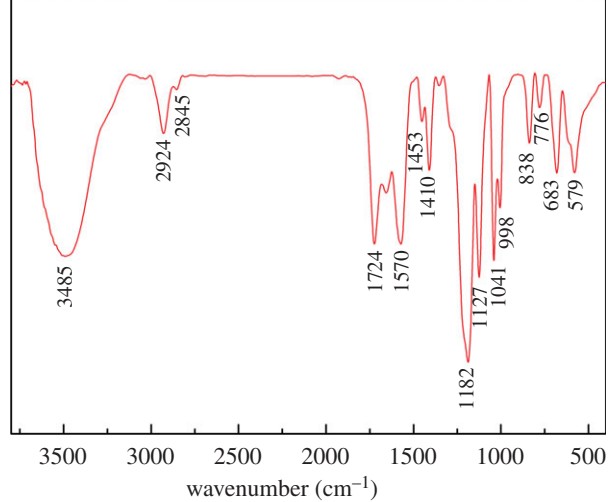

**Figure 1.** Synthetic formula of P(SS-*co*-AA-*co*-DMDAAC).

**Figure 2.** FTIR spectrum of P(SS-*co*-AA-*co*-DMDAAC).

### 2.3.8. CWS stability

The static stability of CWS was analysed by using a TurbiscanLab stability analyser (Formulation Company, France). Before measurements, CWS samples were stirred for 10 min and then filled into the sample bottle to a certain height. Afterwards, the bottle was transferred to the analyser and scanned every 10 min over a period of 5 h. The coefficient Turbiscan stability index (TSI) was used to evaluate the stability of CWS [25,26]. A higher TSI value means poorer static stability.

# 3. Results and discussion

## 3.1. Characterization of P(SS-*co*-AA-*co*-DMDAAC)

In this work, considering the tremendous dosage of dispersant in the CWS industry, free radical polymerization, which is recognized as a proven, economic and efficient technology, was employed. The polymeric formula is shown in figure 1.

FTIR was used to characterize the chemical structure of P(SS-*co*-AA-*co*-DMDAAC). In figure 2, the absorption peak located at 3485 $cm^{-1}$ is due to the stretching vibration of –OH groups. The absorption peaks at 2924 and 2825 $cm^{-1}$ can be attributed to the vibration of -CH$_2$- which connected to N$^+$ groups. The absorption peak at 1724 $cm^{-1}$ belongs to the C=O in carboxyl groups. The peaks at 1570 and 1453 $cm^{-1}$ are assigned to the characteristic absorption of benzene ring. The presence of benzene

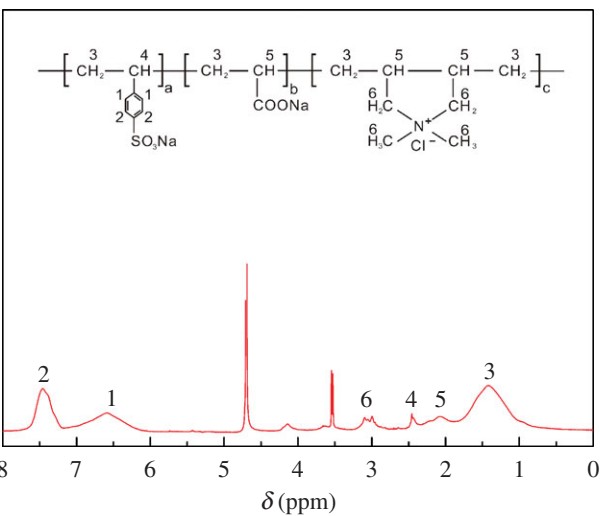

**Figure 3.** $^1$H NMR spectrum of P(SS-*co*-AA-*co*-DMDAAC).

groups can also be confirmed by the clearly observed absorption peaks at 650–900 cm$^{-1}$. The characteristic absorption peaks at 1041 and 1182 cm$^{-1}$ prove the existence of -SO$_3$ groups.

The $^1$H NMR spectrum of P(SS-*co*-AA-*co*-DMDAAC) is shown in figure 3. The chemical shift of P(SS-*co*-AA-*co*-DMDAAC) can be assigned as follows: $\delta = 1.43$, $\delta = 2.08$ and $\delta = 2.45$ are the chemical shift of -CH$_2$- and -CH- protons in backbone of P(SS-*co*-AA-*co*-DMDAAC), respectively. $\delta = 2.98$–3.11 is assigned to the chemical shift of -CH$_3$ and -CH$_2$ connected with N$^+$ group, respectively. $\delta = 6.57$, 7.45 is the chemical shift of proton in the benzene ring. These analyses verified that P(SS-*co*-AA-*co*-DMDAAC) was obtained.

## 3.2. Optimization of synthetic parameters for P(SS-*co*-AA-*co*-DMDAAC)

The feed ratios of monomers could dramatically affect the properties of the copolymer, and further impact the performance of CWS. In industries, CWS with low viscosity is usually desired. Therefore, according to the apparent viscosity, the synthetic conditions of P(SS-*co*-AA-*co*-DMDAAC) were optimized for Yili coal and Yulin coal respectively.

It is well known that both SO$_3^-$ and COO$^-$ functional groups are important for the dispersion performance of dispersants. The content of COO$^-$ and SO$_3^-$ in P(SS-*co*-AA-*co*-DMDAAC) was tailored by changing the feed ratio of AA to SS. As shown in figure 4, with the increasing feed ratio of AA to SS, the viscosities of CWS prepared using Yili coal and Yulin coal exhibit an identical tendency of initially decreasing and then increasing. This tendency can be explained as follows. As both Yulin coal and Yili coal have high ash content, the multivalent cations in ash such as Ca$^{2+}$, Si$^{4+}$ and Al$^{3+}$ which are exposed on the coal surface, make part of the coal surface positively charged [7], although most of the area of the coal surface is negatively charged. The anionic groups in P(SS-*co*-AA-*co*-DMDAAC) such as SO$_3^-$ and COO$^-$ can combine with these positively charged areas by electrostatic attraction, and induce the adsorption of dispersant. Considering the binding ability of COO$^-$ groups to metal ions is much stronger than that of SO$_3^-$ groups, whereas the SO$_3^-$ is more hydrophilic than COO$^-$, COO$^-$ groups had a greater tendency to adsorb onto the coal surface, while the SO$_3^-$ groups extended to water during the adsorption process. As a result, the coal particle surface charge became more negative, and the approach between coal particles was suppressed dramatically due to the enhanced electrostatic repulsion force. Therefore, increasing the feed amount of AA enhanced the adsorption of the dispersant, which is benefit for decreasing the viscosity of CWS. Figure 4 shows that the optimal feed ratio of AA to SS in the synthesis of dispersants for Yili coal and Yulin coal is 1.5:1 and 1:1, respectively. The difference in the optimal ratio may be caused by the difference in the ash content of the two coal samples. When the feed ratios of AA to SS exceeded 1.5:1 or 1:1, the viscosity of CWS increased sharply. This phenomenon may be due to the fact that as the feed amount of AA increased, the relative content of SS units in P(SS-*co*-AA-*co*-DMDAAC) decreased accordingly. The electrostatic repulsive force among coal particles is insufficient to suppress the approach of coal particles. The aggregation of coal particles caused the decrease of free water in the CWS system, and the viscosity of CWS increased.

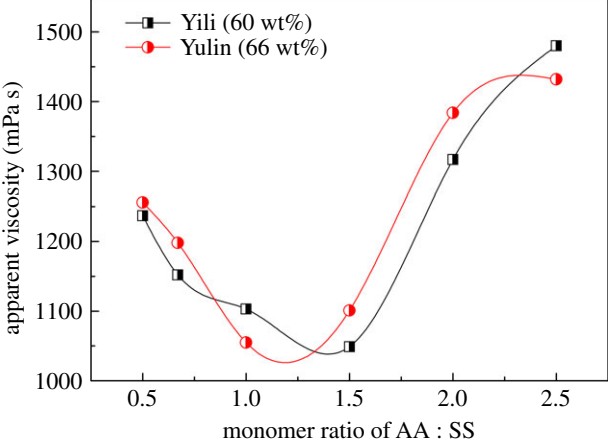

**Figure 4.** Influence of feed ratio of AA to SS on the viscosity of CWS.

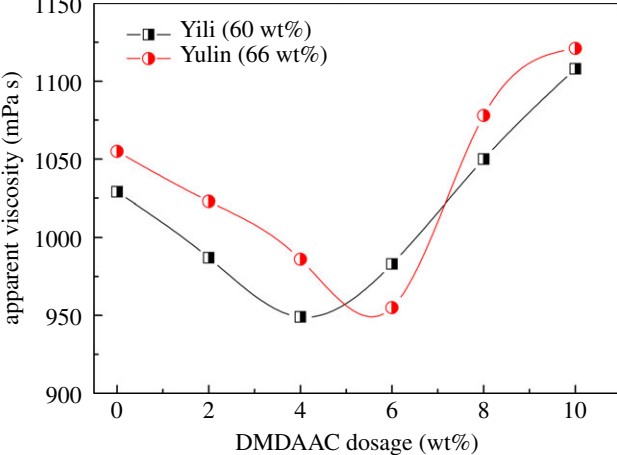

**Figure 5.** Influence of cationic monomer dosage on the viscosity of CWS.

One of the highlights in this work is the introduction of a cationic functional group into the structure of polymer, which is attempted to be used as a dispersant for CWS. Considering that the coal surface is generally negatively charged due to the existence of abundant oxygen groups, the cationic quaternary ammonium groups in P(SS-co-AA-co-DMDAAC) can induce the adsorption of the molecules via electrostatic force. Figure 5 confirms this design, in which the viscosity of CWS decreased with the increase of DMDAAC dosage because of the enhanced adsorption of dispersant. However, when the DMDAAC dosage exceeded the optimal value, the viscosity of CWS increased sharply. This result is reasonable as the excessive cationic groups will neutralize negative charges such as $COO^-$ and $SO_3^-$, and further weaken the electrostatic repulsion force between coal particles. Figure 5 reveals that the optimal dosage of the DMDAAC monomer in the synthesis of dispersants for Yili coal and Yulin coal is around 4.0 wt% and 6.0 wt%, respectively.

## 3.3. Characterization of CWS prepared using P(SS-co-AA-co-DMDAAC) as dispersant

### 3.3.1. Influence of dispersant dosage on the apparent viscosity of CWS

The dosage of dispersant usually has a great impact on the apparent viscosity of CWS. The variation of apparent viscosity against the dispersant dosage is shown in figure 6. As expected, the increase of P(SS-co-AA-co-DMDAAC) dosage, gives the decrease of viscosity initially. This result may be due to the fact that increasing the dispersant dosage leads to more dispersant molecules being adsorbed onto the surface of coal particles and the electrostatic repulsion force between coal particles was enhanced accordingly, which is benefit for the dispersion of coal particles. This assertion can be confirmed by the zeta

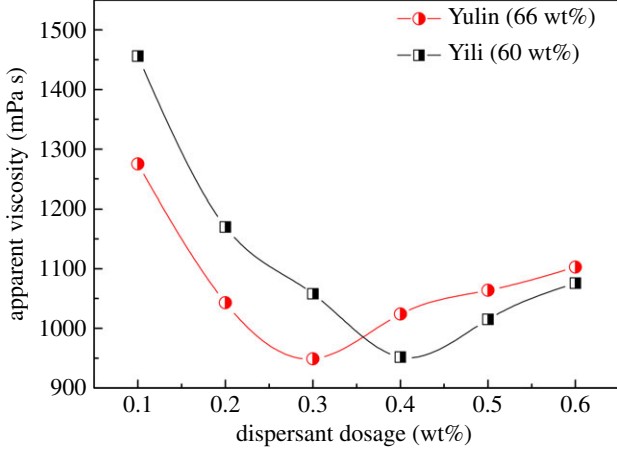

**Figure 6.** Influence of dispersant dosage on the apparent viscosity of CWS.

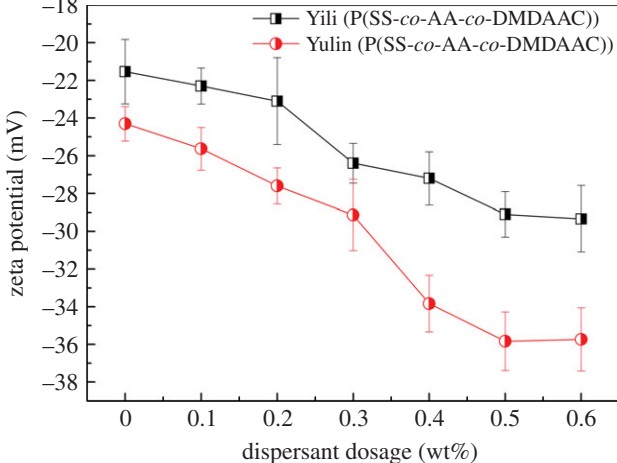

**Figure 7.** Influence of dispersant dosage on the zeta potential of coal particles.

potential measurements, as shown in figure 7. Figure 7 shows that as the dispersant dosage increases, the zeta potential of coal particles becomes more negative and reaches a platform subsequently. Therefore, it can be concluded from figure 6 that the optimal dosage of dispersant for Yulin coal and Yili coal is 0.3 wt % and 0.4 wt%, respectively. Because of its higher ash content, Yili coal has more cations on surface, which combine more anionic groups of the dispersant for adsorption. Accordingly, more dispersant molecules are needed to disperse coal particles. When the dispersant dosages exceed the optimal value, the viscosity of CWS increased to a limited extent. This phenomenon may be due to the fact that the redundant dispersant adsorbed on the coal particles cause the surface become more hydrophilic, and form a too thick hydration film, which reduced the content of free water in the CWS system. Thus, the viscosity of CWS increased. This result confirmed that P(SS-co-AA-co-DMDAAC) can form well-defined adsorption on the surface of coal particles. Anyhow, the optimal dosage of dispersant for Yulin coal and Yili coal are obtained to be 0.3 wt% and 0.4 wt% respectively. Therefore, in the following sections, all the experiments were conducted under this optimized condition.

### 3.3.2. Rheological behaviour of CWS prepared with P(SS-co-AA-co-DMDAAC)

Rheological properties are of great importance for the storage, transportation and burning of CWS. In general, CWS with shear thinning rheological behaviour is desired in industries. Figure 8 shows that the viscosities of two CWS samples were relatively high at a low shear rate and then sharply decreased with the increase in the shear rate, which is consistent with the shear thinning characteristic. For further analysis, two rheological models [7,11,27], Herschel–Bulkley equation (3.1)

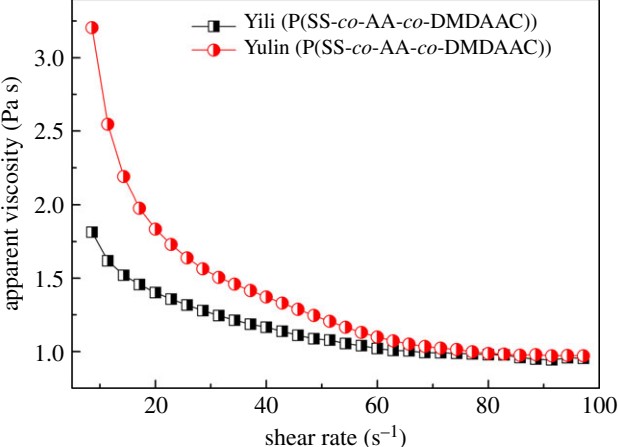

**Figure 8.** Relationship between apparent viscosity and shear rate.

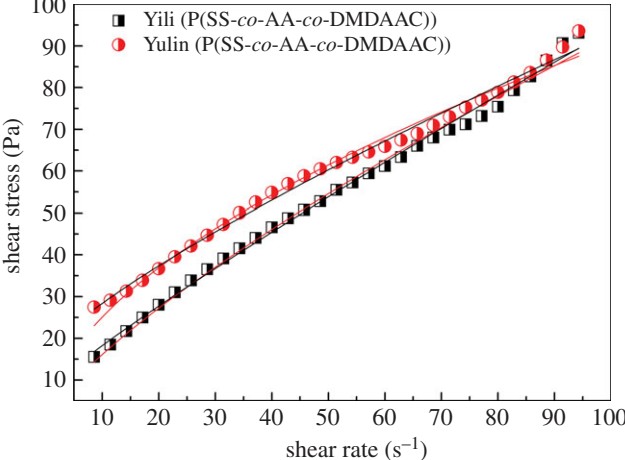

**Figure 9.** Relationship between shear stress and shear rate (Herschel–Bulkley model (black line) and power-law model (red line)).

and power-law equation (3.2), were used to fit the shear stress versus shear rate plot.

$$\tau = \tau_0 + k\gamma^n \tag{3.1}$$

and

$$\tau = k\gamma^n, \tag{3.2}$$

where $\tau$ is the shear stress, $\tau_0$ is the yield stress, $k$ is the fluid consistency index, $\gamma$ is the shear rate and $n$ is the flow behaviour index.

The fitting results are shown in figure 9 and table 2, respectively. The fitting constant $R^2$ indicates that the flow characteristics of two slurries are more in accordance with the Herschel–Bulkley model. The flow behaviour indexes $n$ for both CWS prepared using Yulin coal and Yili coal are less than 1, indicating that slurries are pseudo-plastic fluid, which are in agreement with the results of rheological curve measurements.

### 3.3.3. Static stability of CWS

The static stability of CWS prepared in the presence and absence of dispersant were compared by a stability analyser. The coefficient TSI as a quantitative parameter was used to estimate the stability of CWS [26,28]. As can be seen from figure 10, the TSI of CWS prepared without dispersant is 0.66 and 0.61 for Yili coal and Yulin coal, respectively. After P(SS-co-AA-co-DMDAAC) dispersant was added, the TSI value decreased to 0.35 and 0.22 respectively. Besides, the TSI of CWS prepared using P(SS-co-AA-co-DMDAAC) as dispersant increased slightly after 3600 s, while the CWS without dispersant

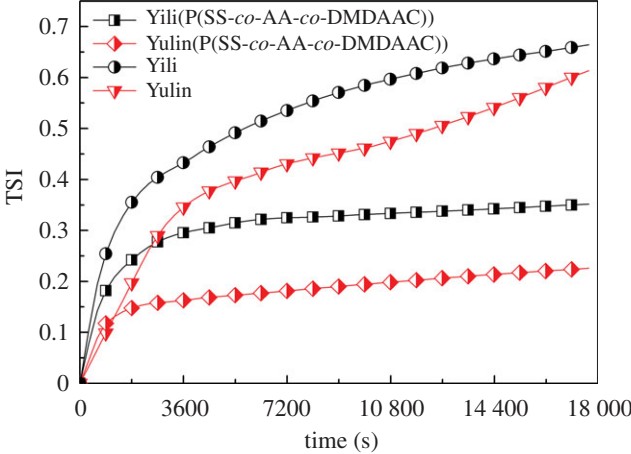

**Figure 10.** Stability of CWS prepared using P(SS-co-AA-co-DMDAAC) as dispersant.

**Table 2.** Fitted parameters of CWS prepared by P(SS-co-AA-co-DMDAAC).

| sample | Herschel–Bulkley | | | | power-law | | |
|--------|------------------|----------|------|-------|-----------|------|-------|
| | $\tau_0$ (Pa) | $k$ (Pa s$^n$) | $n$ | $R^2$ | $k$ (Pa s$^n$) | $n$ | $R^2$ |
| Yili | 7.15 | 1.42 | 0.89 | 0.9949 | 2.86 | 0.76 | 0.9937 |
| Yulin | 16.19 | 1.93 | 0.79 | 0.9933 | 6.96 | 0.56 | 0.9886 |

exhibited a dramatically increasing trend. This result confirms that the stability of CWS is obviously improved by adding P(SS-co-AA-co-DMDAAC).

## 3.4. Comparison of the dispersion performances between P(SS-co-AA-co-DMDAAC) and other dispersants

In this work, a novel CWS dispersant that simultaneously bears anionic and cationic functional groups was designed and synthesized. Based on the above studies, this special structure imparts the dispersant with excellent dispersion and stability performance. Herein, the performance of P(SS-co-AA-co-DMDAAC) was compared with that of NSF and PSS which are widely used CWS dispersants. For practical application, a high concentration CWS is usually desired. However, the viscosity of CWS is inversely proportional to its concentration. In general, the viscosity of CWS should be below 1000 mPa s to ensure the fluidity of slurry. As shown in figure 11, the allowed maximum concentration of CWS prepared using P(SS-co-AA-co-DMDAAC) as dispersant is 60.07 wt% (for Yili coal) and 66.20 wt% (for Yulin coal) respectively, which are obviously higher than that of CWS prepared using NSF or PSS as dispersant. The induction of cationic functional groups accelerated the adsorption of the dispersant. This inference can be verified by the contact angle, adsorption amount and adsorption layer thickness measurements.

It is shown in figure 12 that the contact angle of P(SS-co-AA-co-DMDAAC) aqueous solution against coal surface is lower than that of NSF and PSS. This result indicates that P(SS-co-AA-co-DMDAAC) solution has better wettability to Yulin coal and Yili coal than that of PSS and NSF. The adsorption amount of P(SS-co-AA-co-DMDAAC), PSS and NSF on Yulin coal and Yili coal was displayed in figure 13. It indicated that P(SS-co-AA-co-DMDAAC) adsorbed better than PSS and NSF on Yulin coal and Yili coal. The thickness of adsorption layer formed by dispersant was calculated according to the method provided in previous literatures [24], and the results were shown in figure 14 and table 3. In table 3, the adsorption layer formed by P(SS-co-AA-co-DMDAAC) was thicker than that formed by PSS or NSF. All above measurements verified that P(SS-co-AA-co-DMDAAC) exhibit better wetting and adsorption ability than anionic dispersant PSS and NSF. This beneficial result probably attributes to the introduction of cationic groups, which generate intense electrostatic attraction with coal surface.

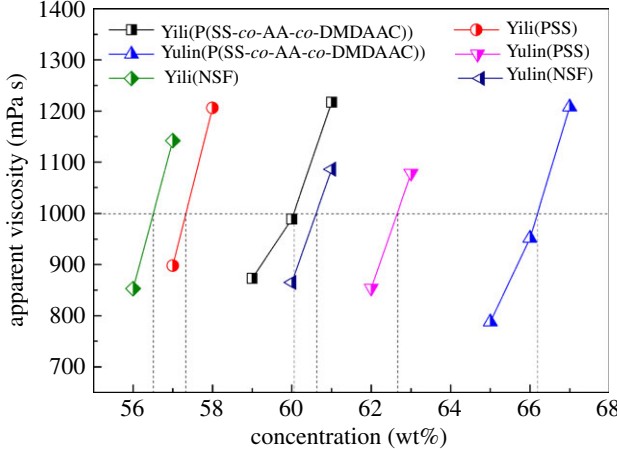

**Figure 11.** Maximum concentration of CWS prepared using P(SS-co-AA-co-DMDAAC), NSF and PSS as dispersant.

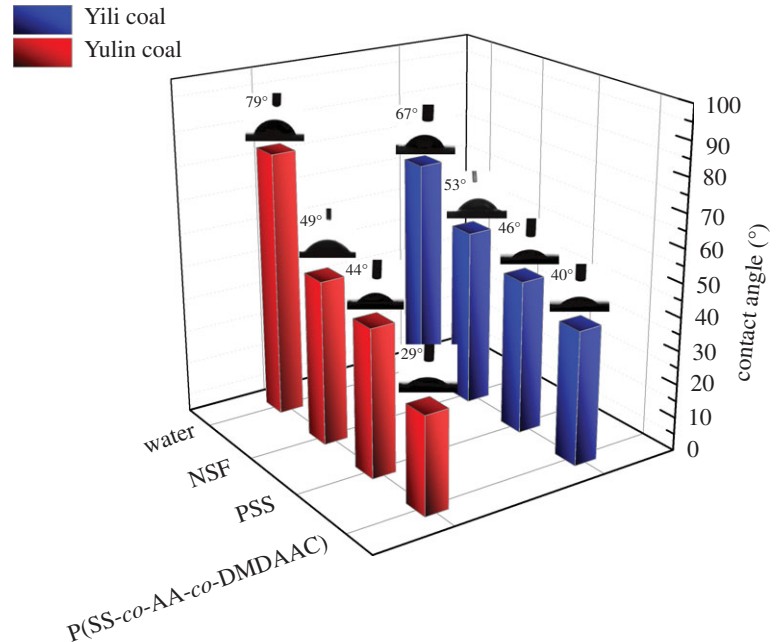

**Figure 12.** Contact angle of water, NSF solution, PSS solution and P(SS-co-AA-co-DMDAAC) solution on Yili coal and Yulin coal.

As a consequence, the adsorbed P(SS-co-AA-co-DMDAAC) further generated strong electrostatic repulsion and steric hindrance between coal particles, which prevented the agglomeration of coal particles. Therefore, CWS prepared using P(SS-co-AA-co-DMDAAC) as dispersant exhibits lower viscosity than that of NSF and PSS.

## 3.5. Adsorption and dispersion mechanism of P(SS-co-AA-co-DMDAAC)

According to the above discussions, the adsorption and dispersion mechanism of P(SS-co-AA-co-DMDAAC) in CWS preparation is speculated, as displayed in figure 15. In our work, P(SS-co-AA-co-DMDAAC) contains three types of functional groups simultaneously, such as $N^+$, $COO^-$ and $SO_3^-$ groups. As previously mentioned, coal surface is generally negatively charged. Therefore, in the adsorption process of P(SS-co-AA-co-DMDAAC), the $N^+$ groups could be anchored to negatively charged area by electrostatic attraction [21,29]. Such anchoring effects are so strong that the P(SS-co-AA-co-DMDAAC) molecules are adsorbed intensely on the coal surface. Moreover, the $COO^-$ groups could combine with metal cations through stable chelate adsorption [16,30], which is also beneficial to adsorption. Consequently, P(SS-co-AA-co-DMDAAC) exhibits superior wetting and adsorbance ability.

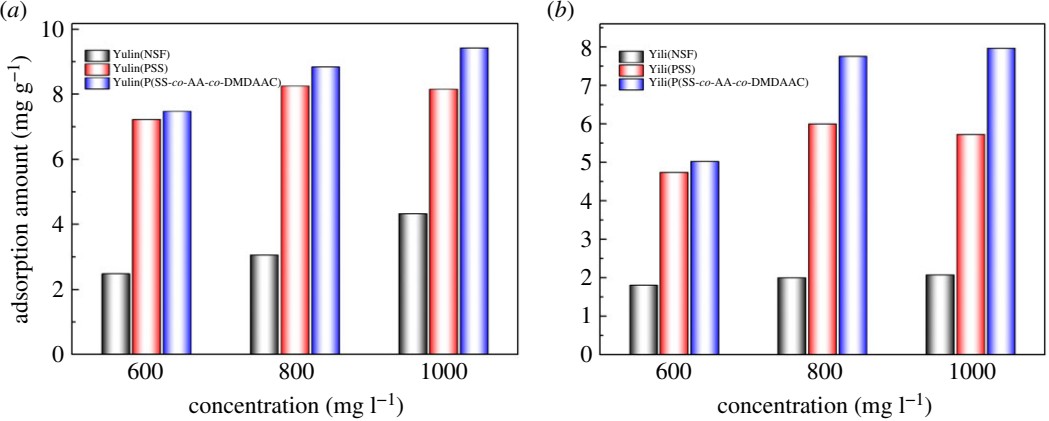

**Figure 13.** Adsorption amount of P(SS-*co*-AA-*co*-DMDAAC), NSF and PSS on Yulin coal (*a*) and Yili coal (*b*).

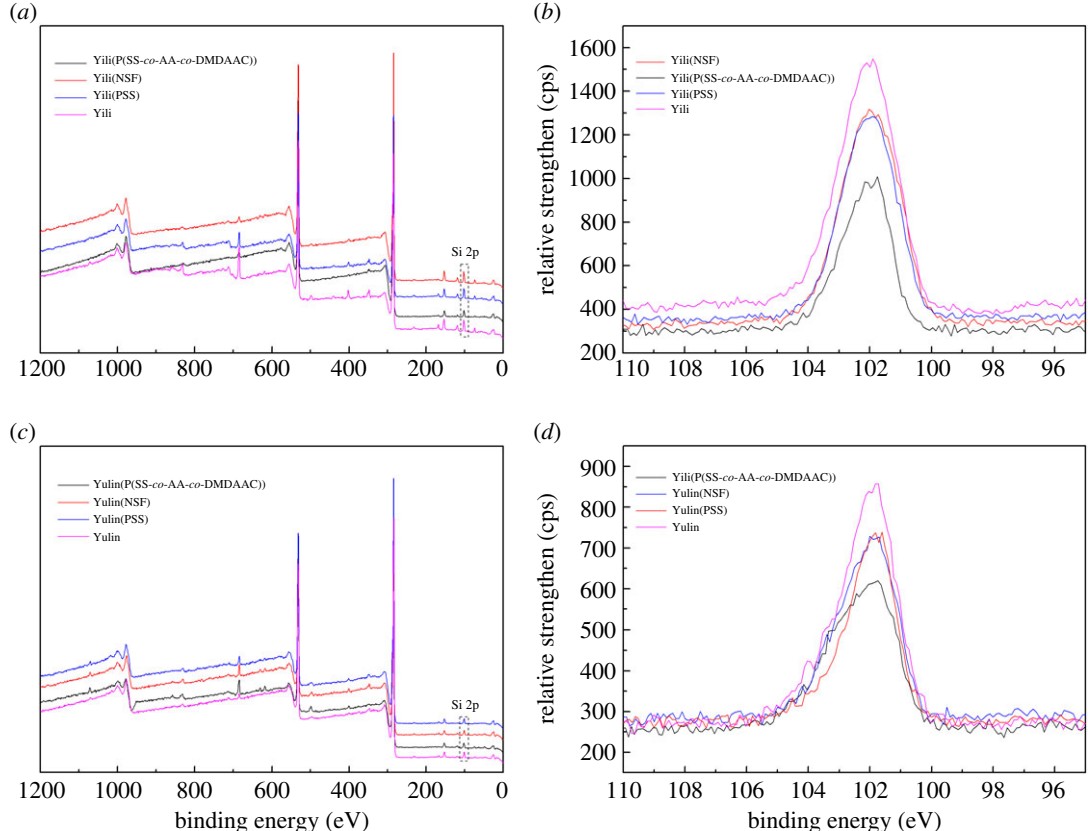

**Figure 14.** XPS spectra of different coal before and after dispersant adsorption (*a*) wide energy spectra of Yili coal (*b*) spectra of Si 2p of Yili coal (*c*) wide energy spectra of Yulin coal (*d*) spectra of Si 2p of Yulin coal.

This speculation is confirmed by the measurements of contact angle, adsorption amount and adsorption layer thickness. Meanwhile, the hydrophilic $SO_3^-$ groups orientate to water, generating electrostatic repulsion among coal particles. A small amount of $COO^-$ groups may also extend into water because of their weak hydrophilicity. As a result, the coal particles surface become more negative charged, the electrostatic repulsion between coal particles is enhanced accordingly. In addition, the $COO^-$ and $SO_3^-$ groups of P(SS-*co*-AA-*co*-DMDAAC) are hydrophilic groups, which induced the formation of hydration film on the coal surface [31]. The hydration film is considered to generate steric hindrance effect [15,32]. Overall, the synergistic effect of wetting, electrostatic repulsion and steric hindrance effectively suppress the agglomeration of coal particles. Therefore, CWS prepared using P(SS-*co*-AA-*co*-DMDAAC) as dispersant exhibits low viscosity and superior stability.

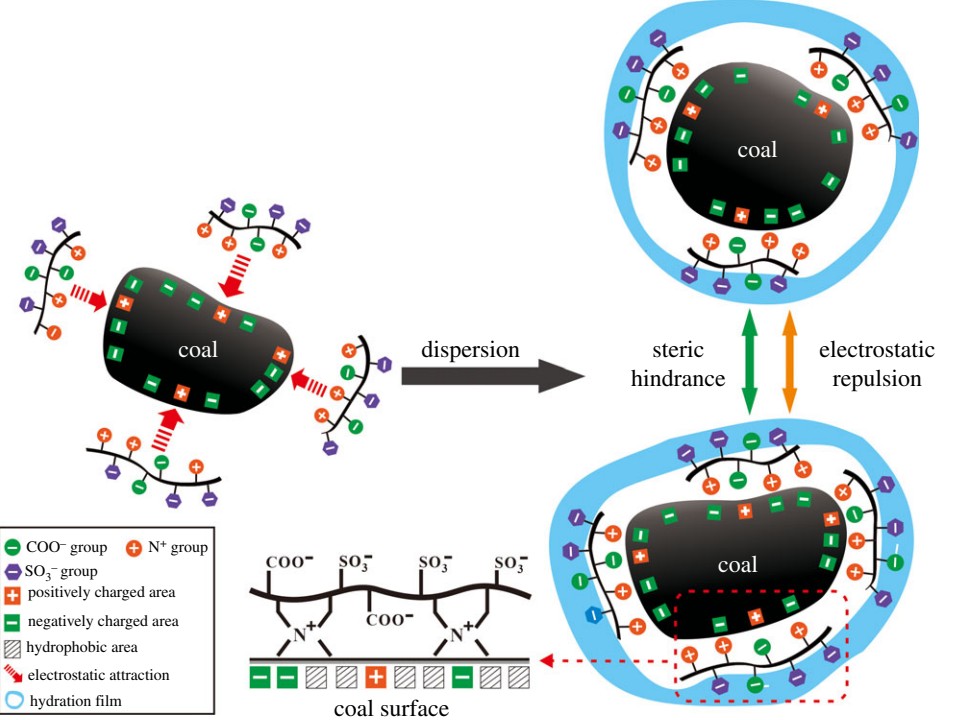

**Figure 15.** Schematic illustration of the adsorption and dispersion mechanism of P(SS-*co*-AA-*co*-DMDAAC) dispersant in CWS.

**Table 3.** Si$_{2p}$ XPS spectra of the raw coal and the coal adsorbed dispersant.

| | Yulin coal | | Yili coal | |
|---|---|---|---|---|
| | area (cps eV) | thickness (nm) | area (cps eV) | thickness (nm) |
| raw | 1382.5 | — | 2302.8 | — |
| NSF | 1154.3 | 0.76 | 2033.9 | 0.52 |
| PSS | 1040.2 | 1.19 | 1976.8 | 0.64 |
| P(SS-*co*-AA-*co*-DMDAAC) | 837.2 | 2.11 | 1489.3 | 1.83 |

# 4. Conclusion

In this paper, a novel amphoteric copolymer named P(SS-*co*-AA-*co*-DMDAAC) was synthesized via free radical polymerization. The FTIR and NMR spectra verified that the copolymer was successfully obtained. The polymeric conditions were optimized with a feed ratio of AA to SS is 1 : 1 and DMDAAC dosage is 4.0 wt% for Yulin coal, while the ratio is 1.5 : 1 and DMDAAC dosage is 6.0 wt% for Yili coal. The dispersion and stability performances of P(SS-*co*-AA-*co*-DMDAAC) used as a dispersant in CWS were evaluated systematically. The optimal dosage of P(SS-*co*-AA-*co*-DMDAAC) for Yulin coal and Yili coal is 0.3 and 0.4 wt% respectively. The CWS prepared using P(SS-*co*-AA-*co*-DMDAAC) shows the rheological characteristics of shear thinning, and more consistent with the Herschel–Bulkley model. Compared with the anionic dispersant NSF and PSS, the amphoteric copolymer P(SS-*co*-AA-*co*-DMDAAC) shows better slurry making performance. The improvements in the application performance mainly attribute to the special amphoteric structure, where the introduction of cationic functional groups enhanced the adsorption of the dispersant.

Data accessibility. The original data are available from the Dryad Digital Repository: https://dx.doi.org/10.5061/dryad.fttdz08qw [33].

Authors' contributions. G.Z., L.D. and D.Y. conceived the original idea and designed the study. J.L. (Jie Luo), Y.L. and C.Z. prepared all samples for analysis. L.D., D.Y., J.L. (Jie Luo) and C.Z. collected and analysed the data. W.Z., J.L. (Junguo Li) and J.Z. participated the analysis of data. L.D., D.Y. and W.Z. wrote the manuscript. All authors gave final approval for publication.

Competing interests. We declare we have no competing interests.

Funding. This study is financially supported by the National Natural Science Foundation of China (grant nos. 21176418, 31670596, 51803111 and 11904220), the Key Research and Development Project of Shaanxi Province of China (grant no. 2020GY-232), the Key Laboratory Scientific Research Project of Shaanxi Provincial Education Department (grant nos. 2013SZS10-K01 and 18JS014), the Project of China Coal Technology & Engineering Corp (grant no. 2019-MS006) and the Natural Science Foundation of Shaanxi Province (grant no. 2019JQ-786).

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
