## [Reviewer comments · Royal Society Open Science]

Review History

RSOS-201480.R0 (Original submission)

Review form: Reviewer 1

Is the manuscript scientifically sound in its present form?

Yes

Are the interpretations and conclusions justified by the results?

Yes

Is the language acceptable?

Yes

Do you have any ethical concerns with this paper?

No

Have you any concerns about statistical analyses in this paper?

No

Recommendation?

Major revision is needed (please make suggestions in comments)

Comments to the Author(s)

1. The language needs a lot of improvements.
2. Contrast test that conducted with common CWS dispersant(s) is needed. This manuscript did not provide enough information.
3. The treatment and testing procedure of the samples for adsorption thickness test was not provided. There was only a brief introduction on the model. Besides, the test results were not discussed. It should be noted that contrast test on adsorption thickness of common dispersant(s) is also needed.
4. The analysis on the mechanism in Section 3.5 is too idealistic. There is lack of data support for the microscopic adsorption behavior of the synthesized dispersant. Besides, wetting dispersion effect and steric hindrance effect are also the key mechanisms for CWS, which were not discussed in this manuscript. It should be noted that steric hindrance is recognized as the most important mechanism, while there is opposite result reported against electrostatic repulsion effect.

Review form: Reviewer 2

Is the manuscript scientifically sound in its present form?

Yes

Are the interpretations and conclusions justified by the results?

Yes

Is the language acceptable?

Yes

Do you have any ethical concerns with this paper?

No

Have you any concerns about statistical analyses in this paper?

No

Recommendation?

Accept with minor revision (please list in comments)

Comments to the Author(s)

This manuscript investigated the preparation of a novel amphoteric copolymer via free radical polymerization and it can improve the dispersion and stability of coal water slurry. The novel dispersant of amphoteric copolymer shows a promising application in industrial. This manuscript could be accepted after minor revisions.

Q1. Why Yili coal and Yulin coal were selected for comparison in the manuscript? What are the characteristics of these two coal samples?

Q2. In abstract, the first letter in the full names of the instruments should be lowercase.

Q3. In 2.3.1, the sample given is a coal sample, but the result of the detection (in Figure 2) is the structure analysis of the dispersant P(SS-co-AA-co-DMDAAC).

Q4. The optimal dosage of dispersant for Yili coal and Yulin coal were different. It is recommended to explain this result in 3.3.1.

Q5. Please check the format of reference 10.

Decision letter (RSOS-201480.R0)

Dear Dr Du:

Title: Synthesis of a novel amphoteric copolymer and its application as a dispersant for coal water slurry preparation
Manuscript ID: RSOS-201480

The editor assigned to your manuscript has now received comments from reviewers. We would like you to revise your paper in accordance with the referee and Subject Editor suggestions which can be found below (not including confidential reports to the Editor). Please note this decision does not guarantee eventual acceptance.

Please submit your revised paper before 15-Oct-2020. Please note that the revision deadline will expire at 00.00am on this date. If we do not hear from you within this time then it will be assumed that the paper has been withdrawn. In exceptional circumstances, extensions may be possible if agreed with the Editorial Office in advance. We do not allow multiple rounds of revision so we urge you to make every effort to fully address all of the comments at this stage. If deemed necessary by the Editors, your manuscript will be sent back to one or more of the original reviewers for assessment. If the original reviewers are not available we may invite new reviewers.

Royal Society of Chemistry

Thomas Graham House
Science Park, Milton Road
Cambridge, CB4 0WF
Royal Society Open Science - Chemistry Editorial Office

On behalf of the Subject Editor Professor Anthony Stace and the Associate Editor Dr Chaohua Cui.

RSC Associate Editor:
Comments to the Author:
(There are no comments.)

RSC Subject Editor:
Comments to the Author:
(There are no comments.)

Reviewers' Comments to Author:
Reviewer: 1

Comments to the Author(s)

1. The language needs a lot of improvements.
2. Contrast test that conducted with common CWS dispersant(s) is needed. This manuscript did not provide enough information.
3. The treatment and testing procedure of the samples for adsorption thickness test was not provided. There was only a brief introduction on the model. Besides, the test results were not discussed. It should be noted that contrast test on adsorption thickness of common dispersant(s) is also needed.
4. The analysis on the mechanism in Section 3.5 is too idealistic. There is lack of data support for the microscopic adsorption behavior of the synthesized dispersant. Besides, wetting dispersion effect and steric hindrance effect are also the key mechanisms for CWS, which were not discussed in this manuscript. It should be noted that steric hindrance is recognized as the most important mechanism, while there is opposite result reported against electrostatic repulsion effect.

Reviewer: 2

Comments to the Author(s)

This manuscript investigated the preparation of a novel amphoteric copolymer via free radical polymerization and it can improve the dispersion and stability of coal water slurry. The novel dispersant of amphoteric copolymer shows a promising application in industrial. This manuscript could be accepted after minor revisions.

- Q1. Why Yili coal and Yulin coal were selected for comparison in the manuscript? What are the characteristics of these two coal samples?
- Q2. In abstract, the first letter in the full names of the instruments should be lowercase.
- Q3. In 2.3.1, the sample given is a coal sample, but the result of the detection (in Figure 2) is the structure analysis of the dispersant P(SS-co-AA-co-DMDAAC).
- Q4. The optimal dosage of dispersant for Yili coal and Yulin coal were different. It is recommended to explain this result in 3.3.1.
- Q5. Please check the format of reference 10.

Author's Response to Decision Letter for (RSOS-201480.R0)

See Appendix A.

RSOS-201480.R1 (Revision)

Review form: Reviewer 2

Is the manuscript scientifically sound in its present form?

Yes

Are the interpretations and conclusions justified by the results?

Yes

Is the language acceptable?

Yes

Do you have any ethical concerns with this paper?

No

Have you any concerns about statistical analyses in this paper?

No

Recommendation?

Accept as is

Comments to the Author(s)

Authors have made appropriate modifications in the revised manuscript and response to the previous comments. It could be published in Royal Society Open Science.

Decision letter (RSOS-201480.R1)

Dear Dr Du:

Title: Synthesis of a novel amphoteric copolymer and its application as a dispersant for coal water slurry preparation

Manuscript ID: RSOS-201480.R1

It is a pleasure to accept your manuscript in its current form for publication in Royal Society Open Science. The chemistry content of Royal Society Open Science is published in collaboration with the Royal Society of Chemistry.

On behalf of the Subject Editor Professor Anthony Stace and the Associate Editor Professor Chaohua Cui.

RSC Subject Editor:
Comments to the Author:
(There are no comments.)

RSC Associate Editor:
Comments to the Author:
(There are no comments.)

Reviewer(s)' Comments to Author:
Reviewer: 2

Comments to the Author(s)
Authors have made appropriate modifications in the revised manuscript and response to the previous comments. It could be published in Royal Society Open Science.

Appendix A

Response to Reviewers Comments (RSOS-201480)

Dear Editor,

Thank you for your letter and for the reviewers' comments concerning our manuscript entitled "Synthesis of a novel amphoteric copolymer and its application as a dispersant for coal water slurry preparation" (RSOS-201480). Those comments are all valuable for improving our paper. We have studied the comments and revised our manuscript carefully before submitting the final version which we hope to meet with approval. All the revised portions are marked in red in the revised manuscript. The authors' response to the reviewer's comments and the main corrections in the revised manuscript are as following:

Response to the Reviewer A

Comment 1: The language needs a lot of improvements.

Reply: Thanks for your comments and questions. We have checked the language of this manuscript carefully and rewrite the sentences and paragraphs. The changes of language are highlighted in red in the revised manuscript. The language in the manuscript had been polished by a professional institution, and the polished proof is shown in Figure R6.

Comment 2: Contrast test that conducted with common CWS dispersant(s) is needed. This manuscript did not provide enough information.

Reply: Thanks for your comments and questions. We have added a common CWS dispersant naphthalene sulfonic formaldehyde condensate (NSF) into the contrast test. The measurements of maximum concentration, contact angle, adsorption amount and adsorption layer thickness were conducted, and these results and discussions were also added to the revised manuscript.

It is shown in Figure R1 that the CWS prepared using P(SS-co-AA-co-DMDAAC) has a higher maximum concentration than that of PSS and NSF. This may attributes to the amphoteric structure of P(SS-co-AA-co-DMDAAC) which improve the adsorption of dispersant on coal surface. The electrostatic attraction between cationic groups of dispersant and oxygen groups on coal surface, as well as the chelation force between carboxyl groups of dispersant and the multivalent cations on coal surface, induce the P(SS-co-AA-co-DMDAAC) molecules to adsorb to the coal surface firmly. The better adsorption of dispersant resulted in

the enhancement of electrostatic repulsion and steric hindrance, which further improved the viscosity and enhanced maximum concentration of CWS. This inference can be verified by the measurements of contact angle, adsorption amount and adsorption layer thickness.

Figure R1. Maximum concentration of CWS prepared using P(SS-co-AA-co-DMDAAC), NSF and PSS as dispersant

The result of contact angle is displayed in Figure R2. Compared with the contact angle of distilled water, the contact angle of P(SS-co-AA-co-DMDAAC) significantly decreases from 67° to 40° for Yili coal and from 79° to 29° for Yulin coal, which are even lower than the that of NSF solution and PSS solution. It indicates that P(SS-co-AA-co-DMDAAC) solution could wet the coal surface faster than PSS and NSF, exhibiting better wettability to coal surface. It can be explained by the introduction of cationic groups and COO⁻ groups which enhances the mutual attraction between P(SS-co-AA-co-DMDAAC) and coal surface. This speculation is further confirmed by the adsorption amount measurement.

Figure R2. Contact angle of water, NSF solution, PSS solution and P(SS-co-AA-co-DMDAAC) solution on Yili coal and Yulin coal

The adsorption amount of NSF and PSS on the surface of Yulin coal and Yili coal was analyzed by using an ultraviolet spectrometer. The ultraviolet absorbance of dispersant solution was measured to build the calibration curve. The calibration curve of P(SS-co-AA-co-DMDAAC), PSS and NSF were displayed in Figure R3. Then 0.5 g blended coal was mixed with 20mL dispersant solution and vibrated in water bath at 25 °C for 5 h. The suspension was centrifuged at 8000 rpm for 10 min, and the supernatant was filtered. Then the absorbance of supernatant was measured and used to calculate the residue concentration of dispersant in supernatant. The adsorption amount of dispersant on coal surface was calculated by following equation.

$$\Gamma = \frac{(C_o - C_r + C_{blank}) \times V}{m} \quad (1)$$

where Γ ($\text{mg}\cdot\text{g}^{-1}$) is adsorbed dispersant on unit mass coal sample, C_o ($\text{mg}\cdot\text{mL}^{-1}$) is original mass concentration of dispersant solution, C_r ($\text{mg}\cdot\text{mL}^{-1}$) is residue mass concentration of dispersant solution after adsorption, C_{blank} ($\text{mg}\cdot\text{mL}^{-1}$) is the mass concentration of blank sample, V (mL) is the volume of dispersant solution, m (g) is the mass weight of coal sample.

Figure R3. Calibration curves of the dispersant concentration to ultraviolet absorbance

The adsorption amount of P(SS-co-AA-co-DMDAAC), PSS and NSF on Yili coal and Yulin coal was calculated and displayed in Figure R4. It showed that the adsorption amount of P(SS-co-AA-co-DMDAAC) on Yulin coal and Yili coal was higher than that of PSS and NSF at the same concentration. Compared with the anionic polycarboxylate dispersant PSS and NSF, the introduction of cationic groups effectively enhanced the adsorption of P(SS-co-AA-co-DMDAAC), especially for Yili coal which has lower coal rank and more hydrophilic surface. This may attributes to the abundant oxygen groups on the surface of coal which could combine with cationic groups of dispersant. Besides, the carboxyl groups which chelate

with metal cations on coal surface can also improve the adsorption of dispersant. Consequently, the increase of adsorption amount of dispersant is beneficial for the enhancement of zeta potential and the formation of hydration film, which further generate stronger electrostatic repulsion and steric hindrance. Thus, the agglomeration of coal particles is suppressed, making the CWS better dispersed. These results and discussions of dispersant adsorption have been added to section 3.4 of the revised manuscript.

Figure R4. Adsorption amount of P(SS-co-AA-co-DMDAAC), NSF and PSS on Yulin coal (a) and Yili coal (b)

Comment 3: The treatment and testing procedure of the samples for adsorption thickness test was not provided. There was only a brief introduction on the model. Besides, the test results were not discussed. It should be noted that contrast test on adsorption thickness of common dispersant(s) is also needed.

Reply: Thanks for your comments and questions. The details of the adsorption layer thickness tests have been added to section 2.3.4 of the revised manuscript. Besides, PSS and NSF were also tested for contrast. Before the XPS measurements, blended coal sample was mixed with dispersant and distilled water and then vibrated at 25 °C in water bath for 5 h. Then the suspension was filtered to remove water and dispersant that did not been adsorbed. After that, the coal sample was dried at 105 °C to constant weight and then attached to the sample plate for XPS analysis. Due to the fact that coal has Si element while NSF, PSS and P(SS-co-AA-co-DMDAAC) has no Si element, Si 2p was selected as the character element for the calculation of adsorption layer thickness. The photoelectron intensity of Si 2p will decay after passing through the adsorption layer. When the dispersant is adsorbed on the coal surface, the Si peak obtained belongs to the Si atoms under the adsorption layer. Thus, the

decay of photoelectron intensity will be used to evaluate the thickness of layer. The thickness of adsorption layer was calculated by the semi-rational equations listed below.

$$d = -\ln(I_d/I_0) * \lambda(E_k) \tag{2}$$

$$\lambda(E_k)=49E_k^{-2}+0.11(E_k)^{1/2} \tag{3}$$

$$E_k=h\nu-E_b \tag{4}$$

where I_d is the intensity of the photoelectron transmitted through the adsorption layer, I_0 is the incident photoelectron intensity, d is the thickness of adsorption layer (nm), $\lambda(E_k)$ is the average depth (nm) at which the light electron escaped, E_k is the light electron kinetic energy and E_b is the atomic binding energy of Si. The value of I_0 and I_d can be obtained according to the area of the Si 2p photoelectron in the XPS spectrum. The XPS analysis results are shown in Figure R5, and the thickness of adsorption layer formed by dispersants are listed in Table R1.

Figure R5. XPS spectra of different coal before and after dispersant adsorption (a) wide energy spectra of Yili coal (b) spectra of Si 2p of Yili coal (c) wide energy spectra of Yulin coal (d) spectra of Si 2p of Yulin coal

Table R1. Si_{2p} XPS spectra of the raw coal and the coal adsorbed dispersant

	Yulin Coal		Yili Coal	
	Area (cps eV)	Thickness(nm)	Area (cps eV)	Thickness(nm)
Raw	1382.5	-	2302.8	-
NSF	1154.3	0.76	2033.9	0.52
PSS	1040.2	1.19	1976.8	0.64
P(SS-co-AA-co-DMDAAC)	837.2	2.11	1489.3	1.83

It is shown in Figure R5 and Table R1 that the integral area of Si 2p peak decreased after the adsorption of dispersant, and area of the coal adsorbed P(SS-co-AA-co-DMDAAC) has the minimum value. The thickness of adsorption layer ranges from 0.76 to 2.11 nm for Yulin coal while it ranges from 0.52 to 1.83 nm for Yili coal. It increases in the order NSF < PSS < P(SS-co-AA-co-DMDAAC). This is consistent with the result of adsorption amount test, verifying that P(SS-co-AA-co-DMDAAC) exhibit better adsorb ability than PSS and NSF no matter on Yulin coal or Yili coal.

Comment 4: The analysis on the mechanism in Section 3.5 is too idealistic. There is lack of data support for the microscopic adsorption behavior of the synthesized dispersant. Besides, wetting dispersion effect and steric hindrance effect are also the key mechanisms for CWS, which were not discussed in this manuscript. It should be noted that steric hindrance is recognized as the most important mechanism, while there is opposite result reported against electrostatic repulsion effect.

Reply: Thanks for your comments and questions. As you mentioned in comments, the wetting effect and steric hindrance effect are also important mechanism which significantly affect dispersion. In order to provide detailed information about the adsorption, the contact angle, adsorption amount and adsorption layer thickness measurements were conducted to evaluate wetting effect. The results showed that the amphoteric structure effectively improved the adsorption of P(SS-co-AA-co-DMDAAC) as compared to anionic dispersants such as NSF and PSS. In the adsorption process of P(SS-co-AA-co-DMDAAC), the N⁺ groups could be anchored to negatively charged area by electrostatic attraction, and meanwhile, the COO⁻ groups could combine with cations through stable chelate adsorption. Such interactions make the P(SS-co-AA-co-DMDAAC) molecules being adsorbed intensely on the coal surface, which are beneficial to wetting. As a result, the P(SS-co-AA-co-DMDAAC) exhibit lower contact angle, larger adsorption amount and thicker adsorption layer as compare to NSF and PSS.

In addition, the COO^- and SO_3^- groups of P(SS-co-AA-co-DMDAAC) which are hydrophilic, could combine with the water molecules which further form hydration film on the outside of the dispersant-coal composite particles [1]. This hydration film is considered to generate steric hindrance among particles [2,3]. Due to the synergistic effect of electrostatic repulsion, steric hindrance and wetting dispersion effect, the agglomeration of coal particles is suppressed. As the consequence, CWS prepared using P(SS-co-AA-co-DMDAAC) as dispersant exhibits low viscosity and superior stability. The discussions of wetting and steric hindrance were added to the section 3.5 in revised manuscript. Besides, the figure of schematic illustration of adsorption and dispersion mechanism was modified.

Response to the Reviewer B

Comment 1: Why Yili coal and Yulin coal were selected for comparison in the manuscript? What are the characteristics of these two coal samples?

Reply: Thanks for your comments and questions. The structure of the copolymer P(SS-co-AA-co-DMDAAC) was designed according to the surface characteristics of low-rank coal. However, the composition of coal is complex and the property differences still exist among different low-rank coal. These property differences can also affect the optimal composition and adsorption ability of dispersant. Thus, in order to reliably evaluate the performances of P(SS-co-AA-co-DMDAAC) for used as a dispersant and its adaptive ability to different types of low-rank coal, we chose Yili coal and Yulin coal. The coal rank of Yulin coal is relatively higher than that of Yili coal. According to the proximate and ultimate analysis result, there are more multivalent cations and hydrophilic oxygen groups on the surface of Yili coal than that of Yulin coal. Thus, the surface of Yili coal is more hydrophilic than the surface of Yulin coal, which is also verified by the moisture content and contact angle.

Comment 2: In abstract, the first letter in the full names of the instruments should be lowercase.

Reply: Thanks for your comments and questions. The appropriate changes have been made in the revised manuscript and highlighted in red.

Comment 3: In 2.3.1, the sample given is a coal sample, but the result of the detection (in Figure 2) is the structure analysis of the dispersant P(SS-co-AA-co-DMDAAC).

Reply: We apologise for the clerical error in the manuscript. In 2.3.1, the sample prepared for FTIR test consists of 5 mg P(SS-co-AA-co-DMDAAC) and 500 mg KBr. The appropriate change has been made in the revised manuscript and marked in red.

Comment 4: The optimal dosage of dispersant for Yili coal and Yulin coal were different. It is recommended to explain this result in 3.3.1.

Reply: Thanks for your comments and questions. We have added the explanation of this result in 3.3.1. The difference of optimal dispersant dosage for Yulin coal and Yili coal mainly attributes to the surface characteristics. Compared with Yulin coal, Yili coal has higher ash content, which means that more cations like Ca^{2+} , Mg^{2+} exists on the surface of Yili coal. During the adsorption process of P(SS-co-AA-co-DMDAAC), more anionic groups of dispersant were combined with cations on the surface of Yili coal, and accordingly, anionic groups which orient to the water and generate electrostatic repulsion decreased. Therefore, more dispersant molecules are required to maintain the coal particles dispersed.

Comment 5: Please check the format of reference 10.

Reply: Thanks for your comments and questions. The format and content of all references has been carefully checked and corrected. The changes are marked in red in the revised manuscript.

Certificate of English Editing

To whom it may concern:

This memo certifies that one of our clients has contracted our academic editing service for the following file.

Title of the paper:

Synthesis of a novel amphoteric copolymer and its application as a dispersant for coal water slurry preparation

Date of the review:

10/10/20 (MM/DD/YY)

The English review was conducted using a two-stage process, in which a junior editor first reviewed the file, and then a senior editor conducted a final and more thorough review. All of our editors are native English-speaking professionals.

Documents receiving this certification should be English-ready for publication; however, the author has the ability to accept or reject our suggestions and changes.

We would like to emphasize that our service targets grammar and language edits. We do not rewrite the documents from scratch. If you are dissatisfied with specific revisions, please contact service@essaystar.com.

Essaystar Group

+1-208-975-4235

EssayStar, 93 S Jackson St, Seattle, WA 98104

Figure R6. Certificate of English Editing for the Manuscript

Reference

1. Zhang K, Jin L, Cao Q. 2016 Evaluation of modified used engine oil acting as a dispersant for concentrated coal – water slurry. *FUEL* **175**, 202–209. (doi:10.1016/j.fuel.2016.02.026)
2. Zhang GH, Zhu N, Li YB, Zhu JF, Jia YR, Ge L. 2017 Influence of side-chain structure of polycarboxylate dispersant on the performance of coal water slurry. *Fuel Process. Technol.* **161**, 1–7. (doi:10.1016/j.fuproc.2017.03.005)

3. Zhou M, Kong Q, Pan B, Qiu X, Yang D, Lou H. 2010 Evaluation of treated black liquor used as dispersant of concentrated coal-water slurry. *Fuel* **89**, 716–723. (doi:10.1016/j.fuel.2009.09.015)